# Effect of Gut Microbiota-Derived Metabolites on Immune Checkpoint Inhibitor Therapy: Enemy or Friend?

**DOI:** 10.3390/molecules27154799

**Published:** 2022-07-27

**Authors:** Haobin Zhao, Di Wang, Zhifu Zhang, Junfang Xian, Xiaosu Bai

**Affiliations:** 1Department of General Practice, People’s Hospital of Longhua, Shenzhen 518109, China; 635274465@mail.nwpu.edu.cn (H.Z.); azhi18607@163.com (Z.Z.); xianjunfang2022@163.com (J.X.); 2Department of Gastroenterology, People’s Hospital of Longhua, Shenzhen 518109, China; diwang_cpu@163.com

**Keywords:** gut microbiota, microbial metabolites, immune checkpoint inhibitor, host immunity, SCFAs

## Abstract

The human gut is inhabited by hundreds of billions of commensal microbiota that collectively produce thousands of small molecules and metabolites with local and systemic effects on the physiology of the host. Much evidence from preclinical to clinical studies has gradually confirmed that the gut microbiota can regulate anti-tumor immunity and affect the efficacy of cancer immune checkpoint inhibitors (ICIs) therapy. In particular, one of the main modes of gut microbiota regulating anti-tumor immunity is through metabolites, which are small molecules that can be transported in the body and act on local and systemic anti-tumor immune responses to promote ICIs immunotherapy efficacy. We discuss the functions of microbial metabolites in humans, focusing on the effects and mechanisms of microbial metabolites on immunotherapy, and analyze their potential applications as immune adjuvants and therapeutic targets to regulate immunity and enhance ICIs. In summary, this review provides the basis for the rational design of microbiota and microbial metabolite-based strategies of enhancing ICIs.

## 1. Introduction

The occurrence and development of cancer are closely related to immune evasion. Improving the immune microenvironment and immune surveillance is of great significance for cancer treatment. In recent years, the advent of immune checkpoint inhibitors (ICIs) has comprehensively reformed the cancer treatment program and effectively improved the survival rate and quality of life of cancer patients. Immune checkpoints are a series of inhibitory pathways for immune cells to regulate and control the persistence of immune response in the process of maintaining immune tolerance. ICIs are known to promote immune-mediated clearance of tumor cells by dampening these inhibitory immune responses. At present, there are three types of ICIs that have been approved for clinical use, including monoclonal antibodies targeting programmed death-1(PD-1), programmed death-ligand 1(PD-L1), and cytotoxic T lymphocyte-associated antigen-4(CTLA-4) [1]. Although ICIs has achieved considerable clinical results, the objective response rates (ORRs) of patients is less than 30% [2,3]. The development of agents that improve the efficacy of ICIs is imminent. The gut microbiota produces a large number of small molecules and metabolites, which play an important role in a variety of physiological processes, including metabolism, endocrine, immunity, etc. [4]. Accumulating evidence supports a role for the gut microbiota in ICIs response in preclinical and clinical studies [5,6]. Not only that, the gut microbiota is also associated with CAR-T cell therapy efficacy [7]. The gut microbiota can improve the effect of immunotherapy by regulating innate immunity, adaptive immunity and immunogenicity of tumor antigen, so as to reshape the tumor microenvironment [8]. Even better, the gut microbiota can not only regulate intestinal immunity, but also shape the host’s immune system through circulation, systemic metabolism and immune regulation. This is due to cytokines and metabolites derived from the gut microbiota. Gut microbial metabolites have a variety of functions, such as regulating the balance of gut microbiota, protecting the intestinal mucosal barrier, maintaining endocrine homeostasis, controlling intestinal immunity, etc. [9]. However, there are few studies on the effects of metabolites derived from gut microbiota on immunotherapy, and the mechanism is still poorly understood.

Here, we reviewed recent studies on the gut microbiota in relation to immunity, summarized the functions of microbial metabolites, and focused on the effects and mechanisms on ICIs. Finally, we explored strategies based on gut microbiota-derived metabolites to improve the efficacy of ICIs.

## 2. Gut Microbial Metabolites and Cancer Immunotherapy

The tumor microenvironment is effectively the soil where tumors grow and the accomplice of immune evasion. The purpose of tumor immunotherapy is to activate or enhance tumor-specific cytotoxic T lymphocytes (CTLs), induce their entry into the tumor microenvironment and exert immune activity to recognize and eliminate cancer cells. Improving the tumor microenvironment may serve as an effective therapeutic strategy for cancer [10]. There are several strategies to achieve this. CD8^+^ T cells are crucial in tumor immunotherapy. Natural killer (NK) cells are also considered as the key effector cells in tumor immunotherapy. Enhancing effector cell cytotoxicity and promoting tumor infiltration can effectively improve immunotherapy [11]. Activation of specific antigen-presenting cells can attenuate tumor immune escape. Regulatory T cells (Tregs) inhibit the immune system from attacking the host’s tissue, but also provide an opportunity for tumor immune escape [12]. Increasing the production of cytokines such as interferon-γ(IFN-γ), interleukin-2(IL-2) and IL-12 can also achieve the positive effect. In contrast, transforming growth factor-β (TGF-β), prostaglandin E2(PGE2) and IL-10 inhibited the activity of immune cells, which affects the efficacy of tumor immunotherapy [13].

The intestine is an important component not only of the human digestive system but also of the immune system. The human intestine is the largest immune organ of the human body, gathering about 65–80% of immune cells. The gut microbiota regulates local and systemic immunity through metabolites, hormones and cytokines. The cross-talk between gut microbial metabolites and immune cells is the main form of interaction between gut microbiota and immunity, and also an important approach to maintaining immune homeostasis [14]. According to different food sources, metabolites of gut microbiota can be divided into the following categories: short-chain fatty acids (SCFAs), bile acids, choline-derived metabolites, indole derivatives, phenolics, vitamins, polyamines, lipids, etc. [15].

### 2.1. SCFAs

SCFAs are saturated fatty acids with a chain length of one to six carbon atoms. They are the major products of dietary fiber fermentation in the colon [16]. They can accumulate to more than 100 mM in the gut depending on the fiber content of the diet [17]. SCFAs mainly include acetate, propionate and butyrate. The approximate molar ratio is 60:20:20 (acetate:propionate:butyrate). Formate, valerate, and caproate are produced in less amounts [18,19]. SCFAs exert a number of effects locally to improve gut health. They are the most abundant anions in the colon and maintain an acidic environment [20]. SCFAs are essential for colonic mucosal health. They promote the production of mucus and antimicrobial peptides from intestinal cells to repair and protect the integrity of supporting intestinal mucosa [21,22,23]. At the same time, they are the energy reservoir of colonocytes and immune cells, supporting immune cells activation and functional maturation [24]. Butyrate, the most widely studied SCFA, not only maintains the balance of gut microbiota and attenuates intestinal inflammation, but also exhibits inhibitory effects on tumorigenesis and reducing the risk of colorectal cancer [25,26,27,28].

#### 2.1.1. SCFA Enhance Immunotherapy Efficacy

Nivolumab or pembrolizumab was administered to patients in a cohort study of 52 patients with solid tumors for whom fecal and plasma SCFAs concentrations were measured, treatment response status and progression-free survival were recorded and analyzed. As a result, higher concentrations of fecal SCFAs were detected in the responder group than in the non-responder group. High concentrations of fecal acetate, propionate, butyrate, and valerate were significantly associated with longer progression-free survival [29]. This indicates a significant correlation between the concentration of SCFAs in the gut and the efficacy of anti-PD-1 mAb immunotherapy.

A study of 11 patients with non-small cell lung cancer (NSCLC) who received second-line nivolumab therapy showed that gut microbial metabolites may play an important role in influencing the response to ICIs. Analysis of the changes of microbial metabolites after treatment showed that 2-pentanone and tridecane were significantly associated with early progression, while SCFAs (propionate and butyrate), lysine and nicotinic acid were significantly associated with long-term beneficial effects. 

Additional preclinical and clinical studies have also implicated SCFAs in the association with ICIs. When examining the difference in the gut microbiota between PD-1 blockade therapy responders and non-responders, it was found that compared with responders, the number of SCFA-producing bacteria in non-responders decreased [30,31].

Moreover, valerate and butyrate were also reported to be able to enhance the efficacy of ROR1-specific CAR-T cells in mice pancreatic cancer models. CAR-T cells after butyrate and valerate treatment, not only highly expressed CD25 but also produced more IFN-γ and TNF-α. The treated CAR-T cells exhibited better anti-tumor efficacy, and more IFN-γ^+^ TNF-α^+^ T cells infiltration was found in tumors [32] (Table 1).

#### 2.1.2. SCFAs Regulate Immune Cell Differentiation and Cytokine Production

SCFAs are critical regulators of intestine immune function and have immunomodulatory capacities on a variety of immune cells, such as regulatory T cells (Tregs), macrophages, antigen-presenting cells, Type 3 innate lymphoid cells (ILC3), and B cells [59,60,61]. During active immune responses, SCFAs can promote the generation of T helper cells (Th1 and Th17). In addition, SCFAs support not only Th1 and Th17 responses in the intestine, but also in systemic tissues, such as the spleen and lymph nodes [62,63]. Acetate and propionate were proved to enhance Th1 and Th17 effector T cells (T_eff_) polarization [64]. 

SCFAs also regulate the expression of cytokines of immune cells. Butyrate can escalate the production of IL-22 by ILC3 and CD4^+^ T cells. Moreover, it was proven to promote IFN-γ production and increases the cytotoxicity of CD8^+^ T cells. Valerate has also been shown to enhance the expression of TNF-α and IFN-γ in human CD8^+^ T cells and CAR-T cells [32]. In addition, acetate indirectly stimulated the IL-22 secretion by enhancing the expression of IL-1 receptors in ICL3 cells [60].

#### 2.1.3. Mechanisms of SCFAs

The mechanism by which SCFAs exert immune regulatory effects is dependent on histone deacetylase (HDAC), mTOR, G protein coupled receptors (GPRs) and metabolic regulation (Figure 1). 

In addition to promoting cell differentiation and cytokine secretion, butyrate also significantly increased the intratumoral accumulation of CD8^+^ T cells. The molecular mechanism was that butyrate induced ID2 expression by inhibiting histone deacetylase (HDAC) activity. Then, ID2 promoted anti-tumor immunity of CD8^+^ T cells by regulating IL-12-dependent pathway [65]. Valerate could promote the antitumor activity of CTLs and CAR-T cells by inhibiting class I HDAC enzymes [32]. Acetate is not a HDAC inhibitor, but it can also inhibit HDAC in activated T cells [62]. Butyrate escalates the production of IL-22 in ILC3 and CD4^+^ T cells via GPR41 and inhibiting HDAC [66]. Propionate-induced GPR43 agonism promotes the function of colonic ILC3s and then increases the production of ILC3-derived IL-22 via AKT and STAT3 signaling pathways [67].

SCFAs could regulate T cell function by modulating cellular metabolism through the mTOR pathway. The mTOR activity is essential for the normal differentiation of T_eff_ cells and Treg cells [68]. Generally, mTOR activity promotes T_eff_ cell differentiation at high levels but promotes Foxp3^+^ Treg generation at low levels [62,68]. It was achieved through inducing S6K acetylation downstream of the mTOR pathway. Activation of STAT3 and mTOR pathways in Th1 cells by butyrate promotes IL-10 expression mediated by GPR43 [69].

SCFAs can also be converted into acetyl coenzyme A and participate in the TCA cycle of cell energy metabolism. The TCA cycle produces ATP and metabolic building blocks to provide fuel for multiple cellular activities, such as T cells differentiation [70]. In addition to these, SCFAs can regulate immunity through module function of B cells and antibody secretion. This is controlled through the participation of acetyl CoA in the metabolism of B cells, in concert with the HDAC and GPR regulation of gene expression [70].

### 2.2. Inosine Enhance Efficacy of ICIs

Inosine was mainly derived from the metabolism of adenine moieties of nucleic acid by the gut microbiota. It has the function of regulating the gut microbiota and protecting intestinal mucosa barrier [71]. Researchers isolated a strain of gut microbiota, *Bifidobacterium pseudolongum*, from mice, which could promote the therapeutic efficacy of anti-CTLA-4 mAb and anti-PD-L1 mAb on colon cancer. Colonization of B. pseudolongum in intestine promoted the metabolism of the gut microbiota to produce inosine and increase the content of inosine in serum [72]. Oral administration of inosine improved therapeutic efficacy of anti-CTLA-4 and the effect was CpG-dependent. If CpG was not given concomitantly, inosine would promote tumor growth instead of increasing antitumor immune responses. Inosine also improved the efficacy of anti-CTLA-4 mAb and anti-PD-L1 mAb in mouse models with bladder cancer and small bowel cancer [72].

#### Mechanism of Inosine

Inosine could regulate the Th1 differentiation and increased IFN-γ^+^ CD4^+^ and CD8^+^ T cell infiltration. However, it can promote the differentiation from T cells to Th1 only when exogenous IFN-γ exists, otherwise it will inhibit the Th1 differentiation. Its molecular mechanism is to activate cyclic adenosine monophosphate (cAMP)-response element binding protein (CREB), a known translational enhancer of key Th1 differentiation factors, through adenosine 2A receptor (A2aR)-cAMP-protein kinase A cascade signal [72].

In vitro experiments confirmed that inosine could serve as an alternative substrate to support T_eff_ cell growth and function in the absence of glucose. Furthermore, the supplementation of inosine enhanced the anti-tumor efficacy of immune checkpoint blocking or adoptive T cell metastasis in solid tumors with defects in the metabolism of inosine, reflecting the ability of inosine to alleviate the metabolic restriction imposed by tumors on T cells. The combination of inosine supplementation and anti-PD-L1 mAb therapy increased tumor infiltrating T cells and IFN-γ and TNF-α expression compared with anti-PD-L1 mAb therapy alone. It also increased the percentage of circulating CD8^+^ T cells expressing IFN-γ in tumor draining lymph nodes and spleen. However, when tumor cells are able to compete with T cells for inosine utilization, the beneficial effect of inosine supplementation will be diminished [36] (Figure 2, Table 1).

The adenine moiety of nucleic acids is a major source of inosine. Inosine can protect intestinal mucosa and maintain immune homeostasis. In the deficiency of glucose, inosine can be an energy source for CD8^+^ T cells and promotes IFN-γ release. Through activating A2aR, inosine can promote Th17 cell differentiation.

### 2.3. Trimethylamine Oxide (TMAO)

Foods such as fish, eggs, and meat products contain large amounts of choline or carnitine, which can be metabolized by the gut microbiota to generate trimethylamine (TMA). TMA enters the liver through portal vein circulation and is catalyzed to produce trimethylamine oxide (TMAO) [73]. TMAO is famous for predicting the risk of adverse cardiovascular outcomes. There is a dose-dependent association between plasma TMAO concentration and cardiovascular disease risk in cardiac patients, which was first discovered in 2011 [74]. Moreover, elevated circulating TMAO is associated with an increased risk of colon [75] and prostate cancer [76].

#### TMAO Enhanced Efficacy of Anti-PD-1mAb

In the study on the ability of microbial metabolites to regulate the tumor microenvironment by non-targeted metabonomics, the researchers found that TMAO inhibits tumor growth by activating CD8^+^ T cell-mediated antitumor immunity. Combined intratumoral injection of TMAO and intraperitoneal injection of anti-PD-1 mAb suppressed tumor growth more significantly than injection of anti-PD-1 mAb alone. Intratumoral injection of TMAO significantly promoted the infiltration of CD8^+^ T cells and M1 macrophages and enhanced the function of CD8^+^ T cells. Moreover, elevated levels of IFN-γ and TNF-α were detected in tumors injected with TMAO. After depletion of CD8^+^ T cells from mice using a CD8 neutralizing antibody, the antitumor effect of TMAO was attenuated. Clinical data show that patients with high plasma TMAO concentrations have longer progression-free survival and better response to ICIs than those with low plasma TMAO concentrations. TMAO improves anti-tumor immunity by activating PERK mediated ER stress, triggering Gasdermin-E mediated pyroptosis [37] (Figure 3, Table 1).

Carnitine acid and choline would be metabolized to TMA by the gut microbiota. After entering the liver through portal vein circulation, TMA is converted to TMAO. At the same time, TMAO can increase the tumor infiltration of CD8^+^ T cells and the release of effector cytokines. Through these, TMAO can improve the effect of immunotherapy.

### 2.4. Tryptophan Metabolites

Tryptophan is one of the essential amino acids for the human body and the only one that contains indole structure. Studies have shown that tryptophan could be catabolized by the gut microbiota to generate a variety of metabolites, such as indole, indolelactic acid, indolepropionic acid, indoleacrylic acid, tryptamine, and others. Tryptophan metabolites play important roles in regulating intestinal immunity, protecting the intestinal barrier, controlling hormone secretion, and stimulating gastrointestinal motility [48].

The aryl hydrocarbon receptor (AhR) is a crucial regulator of immune responses, making this receptor an attractive molecular target for immunotherapy. Most tryptophan metabolites can bind to AhR and activate downstream pathways to regulate immune response [77]. Tryptophan (Trp) is converted to Kynurenine (Kyn) after catalysis via 2,3-dioxygenase 1 (IDO1), IDO2, and tryptophan-2,3-dioxygenase (TDO) in several cancers. The Kyn-AhR pathway has proven to be an important way to limit immunotherapy. Despite many setbacks, tumor immunotherapeutic drugs targeting IDO or tryptophan metabolism are still under development [78].

#### Gut Microbiota-Derived Tryptophan Metabolites

In murine pancreatic ductal adenocarcinoma models, indole-derived tryptophan metabolites from *Lactobacillus* suppressed tumor immunity. A2aR signaling in tumor-associated macrophages was activated by tryptophan metabolites, and the release of IFN-γ from infiltrating CD8^+^ T cells to kill tumor cells was inhibited. Upon removal of tryptophan in the diet, tumor growth was suppressed and immunity was enhanced. Conversely, dietary supplementation of indole-derived tryptophan metabolites in turn triggered immunosuppression [39] (Figure 4).

However, other studies provided counter evidence for indole-derived tryptophan metabolites. Indole-3-lactic acid (ILA) not only induces apoptosis of colon cancer cells in vitro but also suppresses intestinal tumorigenesis in mice [41]. ILA metabolized by *Bifidobacterium* in infants is able to promote IL-22 expression during Th17 cell differentiation by acting on the AhR on the surface of T cells. IL-22 plays an important role in maintaining intestinal epithelial cell tight junctions and defending against pathogenic microbial invasion at the intestinal mucosa [44]. Idolyl-3-propionic acid (IPA) was proven to inhibit the growth of breast cancer in vitro and vivo. The effect was exerted through AhR and pregnane X receptor (PXR). In addition, AhR activation and PXR expression were negatively correlated with cancer cell proliferation, tumor stage and grade. In women newly diagnosed with breast cancer, especially stage 0 women, IPA biosynthesis of the gut microbiota was inhibited [45]. Indole-3-carboxaldehyde (3-IAld) could attenuate colitis induced by ICIs in mice, but did not affect therapeutic efficacy. The beneficial activity of 3-IAld was achieved by increasing the intestinal barrier through the AhR/Il-22 axis and controlling inflammation through Treg cells [46] (Figure 4, Table 1).

Tryptophan metabolites can protect intestinal mucosa, regulate intestinal immunity and control hormone secretion. Most tryptophan metabolites can activate AHR signaling and suppress immune responses. IPA and ILA can inhibit tumorigenesis and growth.

### 2.5. Secondary Bile Acids

Primary bile acids are synthesized in the liver and contribute to fat metabolism in the enterohepatic circulation. Primary bile acids are converted to secondary bile acids by the gut microbiota metabolism. Bile acids activate bile acids receptors primarily through G protein-coupled bile acid receptor 1 (GPBAR1 or TGR5) and the nuclear hormone receptor farnesoid X receptor (NR1H4 or FXR). Other nuclear hormone receptors targeted by bile acids include pregnane X receptor, vitamin D receptor, constitutive androstane receptor, and membrane bound receptors such as sphingosine-1-phosphate receptor 2 and cholinergic receptor muscarinic 2. Through these receptors and downstream signals, bile acids can mediate enteroendocrine hormone secretion, maintaining immune and metabolic homeostasis [79,80].

Primary bile acids can promote CXCL16 expression in liver sinusoidal endothelial cells in mice. This leads to the accumulation of CXC chemokine receptor 6^+^ natural killer T (NKT) cells in the liver, which induce increased IFN-γ secretion and liver tumorigenesis. In this case, the gut microbiota metabolizes primary bile acids to secondary bile acids, protecting the liver from tumorigenesis [81].

The metabolite of lithocholic acid (LCA), a secondary bile acid generated by the gut microbiota, is known as 3-oxoLCA. Some bacteria convert LCA to 3-oxoLCA, others convert 3-oxoLCA to isoLCA. Similar to 3-oxoLCA, isoLCA suppressed Th17 cell differentiation by inhibiting retinoic acid receptor-related orphan nuclear receptor-γt (ROR-γt), a key Th17-cell-promoting transcription factor. The levels of these bile acids were inversely correlated with the expression of Th17-cell-associated genes [51] (Figure 5, Table 1).

Secondary bile acids are metabolites of primary bile acids synthesized by the liver. They can regulate metabolic and immune homeostasis and control hormone secretion. Secondary bile acids can prevent hepatic tumorigenesis by suppressing hepatic NKT cell initiated inflammatory responses via inhibiting CXCL16 expression. The metabolite 3-oxoLCA may inhibit Th17 cell differentiation by suppressing RORγt.

### 2.6. Other Microbial-Derived Molecules

Hippuric acid, a phenolic produced from the metabolism of phenyl or phenolic compounds [82]. A clinical trial on the correlation between plasma metabolites and response to PD-1 blockade therapy in NSCLC patients showed that the combination of hippuric acid, butyrylcarnitine, cysteine, and glutathione disulfide had a high response probability [52]. This may be due to their association with T cell metabolism. In a prospective validation set, the metabolites combination showed a high accuracy in the clinical response of patients, which may serve as predictive biomarkers of the PD-1 blockade therapy for clinical usage [52] (Table 1).

Muropeptides generated by *Enterococci* enhanced anti-PD-L1 mAb immunotherapy in mice. *Enterococci* expressed and secreted a homolog of the NlpC/p60 peptidoglycan hydrolase SagA, which could resolve bacterial cell walls and release muropeptides. Muropeptide exerted the immune activation function through the signal transduction of innate immune sensing protein NOD2, increased the proportion of CD8^+^ T cells in tumor infiltrating lymphocytes and expressing granzyme B. Tumor immune microenvironment was improved and the efficacy of multiple immunotherapeutic mAbs was enhanced [54] (Table 1).

Microbial exopolysaccharide produced by *Lactobacillus delbrueckii subsp. bulgaricus* OLL1073R-1 (EPS-R1) was proven to enhance efficacy of anti-CTLA-4 mAb or anti-PD-1 mAb immunotherapy. In CCL20-expressing tumor bearing mice, EPS-R1 induced CCR6 expression in CD8+ T cells by binding to lysophosphatidic acid receptor on cells. Ingestion of EPS-R1 significantly increased the number of CCR6^+^ population in CD8^+^ tumor infiltrating lymphocytes (TILs). EPS-R1 further augmented the expression of IFN-γ as well as the genes encoding IFNγ-inducible chemokines to enhance T cell function [55] (Table 1).

FimH is an adhesin of type I pili on the surface of Gram-negative bacteria such as *Escherichia coli*. As a ligand of Toll-like receptor 4 (TLR4), FimH can activate mouse and human NK cells by binding to TLR4 [57]. FimH combined with OVA can promote antigen-specific immune activation, promote T cell proliferation and IFN-γ and TNF-α production, increase the tumor infiltration of T_eff_ cells and inhibit the growth of melanoma in mice. FimH in combination with anti-PD-L1 mAb can promote the activation of dendritic cells (DCs) and enhance the effect of anti-PD-L1 mAb on CT26 xenografts in mice [56] (Table 1).

Castalagin, a polyphenol from berry camu-camu. It could alter the gut microbiota composition in mice and enhance antitumor activity and anti-PD-1 response in mice [58]. Oral administration of castalagin enriched for bacteria associated with efficient immunotherapeutic responses (*Ruminococcaceae* and *Alistipes*) and improved the CD8^+^/FOXP3^+^CD4^+^ ratio within the tumor microenvironment. Castalagin preferentially binds to the extracellular membrane of *Ruminococcus bromii* and promotes an anticancer response. Oral administration of either *Ruminococcus* or castalagin alone had no apparent effect in germ free mice, which suggested that the effect of castalagin was gut microbiota-dependent [58] (Table 1).

## 3. Future Perspectives

The small molecules and metabolites derived from probiotics are called postbiotics, and their industrialization in food, health care products, skin care products and other fields is in full swing [83]. In the above, we introduced the important roles of gut microbial metabolites in maintaining metabolic balance, protecting the intestinal mucosa, and promoting immune responses. Gut microbial metabolites enhance the efficacy of immunotherapy mainly by relieving the immunosuppression in the tumor microenvironment and improving the cytotoxicity of T_eff_ cells. Future work will seek to understand how we can utilize gut microbial metabolites in the clinic to benefit more patients from immunotherapy.

### 3.1. Strategies

Firstly, microbial metabolites-based adjuvants for ICIs are in development. SCFAs and other postbiotics are being tested in clinical trials. Regarding microbial-derived tryptophan catabolites, targeting this pathway provides several inhibitors to overcome the immunosuppression [78]. Secondly, microbial metabolites have the potential to be predictive biomarkers of ICIs response. As mentioned above, SCFAs, idolyl-3-propionic acid and TMAO have been trying to serve this role. Third, fecal microbiota transplantation is the transplantation of the gut microbiota from ICIs responders to non-responders. This strategy has been applied in several clinical trials and performed well [5]. The fourth one is synthetically engineered microorganisms. Advances in synthetic biology and genetic engineering technology have enabled the design of microorganisms based on their therapeutic needs [84]. Last but not the least, diet has a great influence on the composition of the gut microbiota. There is plenty of evidence that dietary fiber supplementation in the diet can adjust the composition of the gut microbiota and promote the effectiveness of ICIs. In contrast, a high-fat diet increases the risk of colorectal cancer [15].

### 3.2. Present Problems

Although effects of microbial metabolites have been supported by a lot of research evidence, the controversy still exists. Coutzac’s research revealed the phenomenon that high serum concentrations of butyrate and propionate, respectively, were negatively correlated with ICIs efficacy in two cohorts of patients with cancers treated with ipilimumab. In the subcutaneous tumor model experiment in mice, butyrate supplementation reduced anti-CTLA-4 mAb treatment efficacy [35]. More efforts are required to establish role of microbial metabolites in different kinds of cancer and immunotherapy.

The A2aR is a high affinity receptor that is expressed on various immune cells. It is intriguing that adenosine activates A2aR signaling to manifest as immunosuppression, whereas inosine activates A2aR signaling to enhance immune responses [72,85]. The mechanisms of action of microbial metabolites that modulates immune microenvironment still needs to be more deeply explored.

Fecal microbiota transplant carries the risk of infection with drug-resistant organisms. In contrast, supplementation with microbial metabolites has a prominent safety advantage. However, the use of purified metabolites may upset the microbiota balance. Currently, several clinical trials exploring the relationship between the gut microbiota, microbial-derived metabolites and ICI efficacy or toxicity are ongoing.

## 4. Conclusions

With the emergency of ICIs, the field of cancer treatment has made major breakthroughs. Despite significant improvements in patient survival, resistance and response to ICIs remain as challenges to be overcome. The gut microbiota plays an important role in regulating human immunity. Gut microbial metabolites act in tandem to crosstalk between the host and the microbiota. Due to its immunomodulatory effect, gut microbial metabolites have attracted the interest of many immunotherapy researchers. Although more studies support the claim that short chain fatty acids can promote the effect of immunotherapy, the research evidence of a few counterexamples cannot be ignored. Inosine and TMAO may be effective adjuvants for ICIs, but more clinical trials are needed. There are many kinds of metabolites derived from tryptophan and bile acids, and their performance is different, which should be discussed according to specific types. How can microbial metabolites be exploited to achieve beneficial effects? It is necessary to conduct more in-depth studies to explore the effects of microbial metabolites on different cancers and different immunotherapies. We look forward to seeing more trials focusing on the precise targets and mechanisms of action of microbial metabolites in the future. More microbial metabolites-based immunotherapeutic agents will be applied in cancer immunotherapy. This will contribute to achieving a more accurate and personalized treatment of cancer.

## Figures and Tables

**Figure 1 molecules-27-04799-f001:**
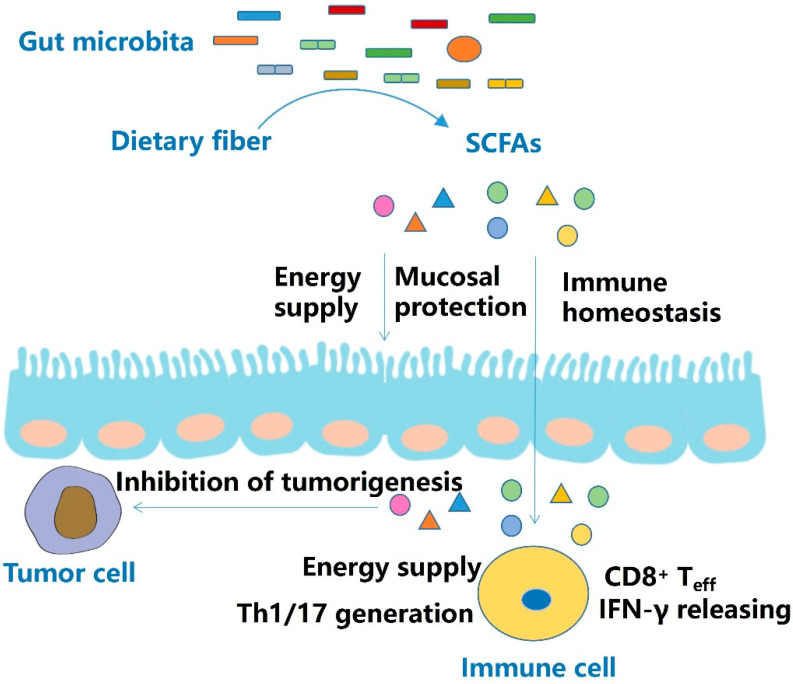
Effect and mechanism of SCFAs. SCFAs are mainly derived from dietary fibers. SCFAs not only provide energy to intestinal cells, but also protect the intestinal mucosal barrier and maintain intestinal immune homeostasis. In addition to suppressing colon tumorigenesis, SCFAs elevate the population of effector T cells (T_eff_), promote cytokine release, and enhance immunotherapeutic efficacy.

**Figure 2 molecules-27-04799-f002:**
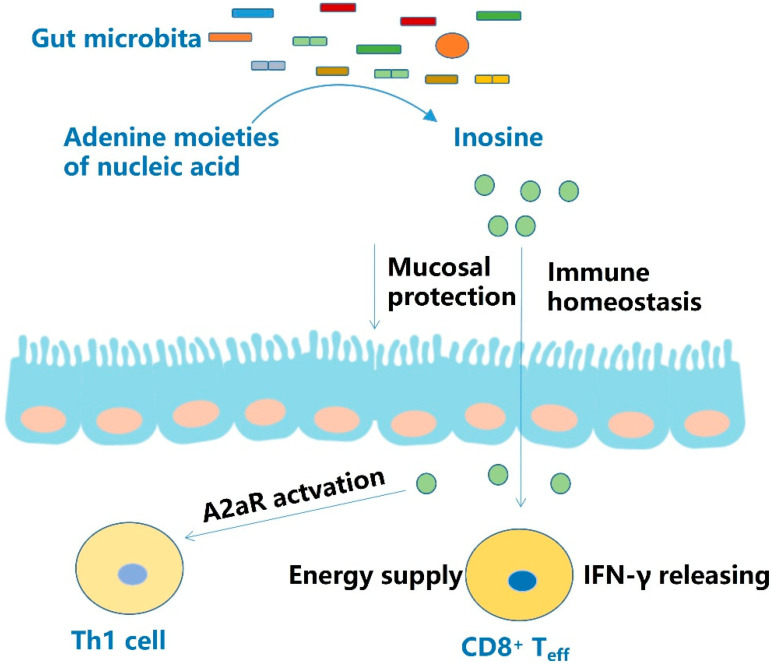
Effect and mechanism of inosine.

**Figure 3 molecules-27-04799-f003:**
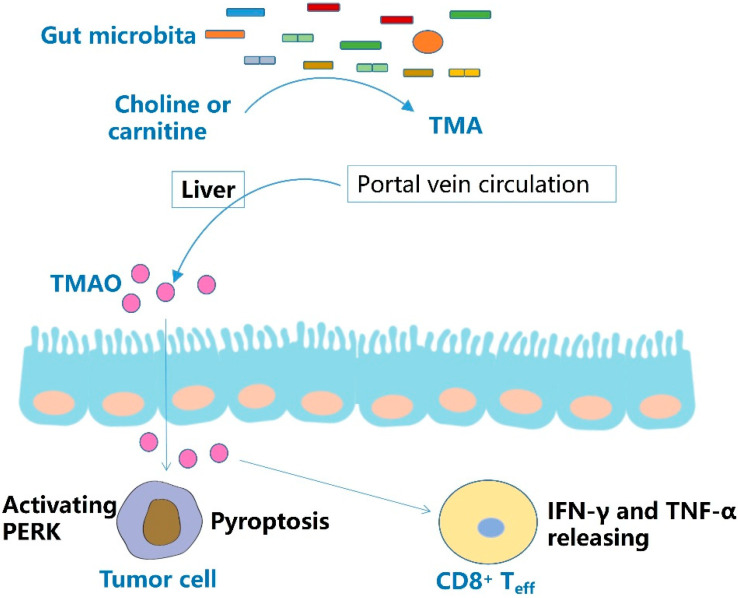
Effect and mechanism of TMAO.

**Figure 4 molecules-27-04799-f004:**
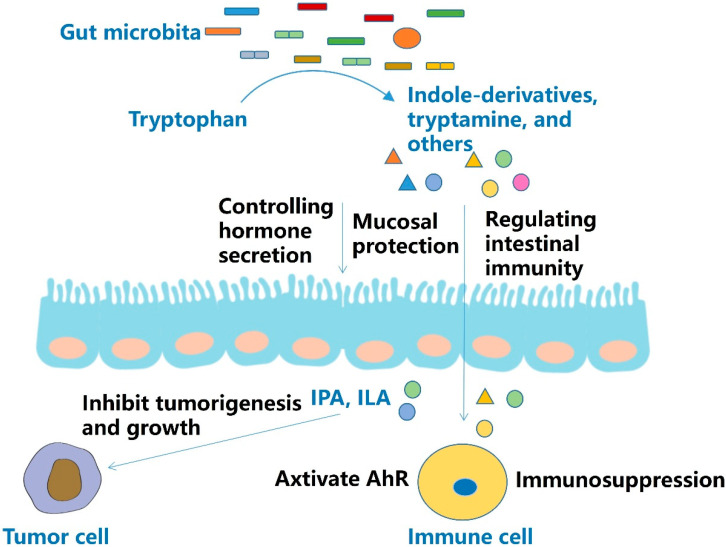
Effect and mechanism of tryptophan metabolites.

**Figure 5 molecules-27-04799-f005:**
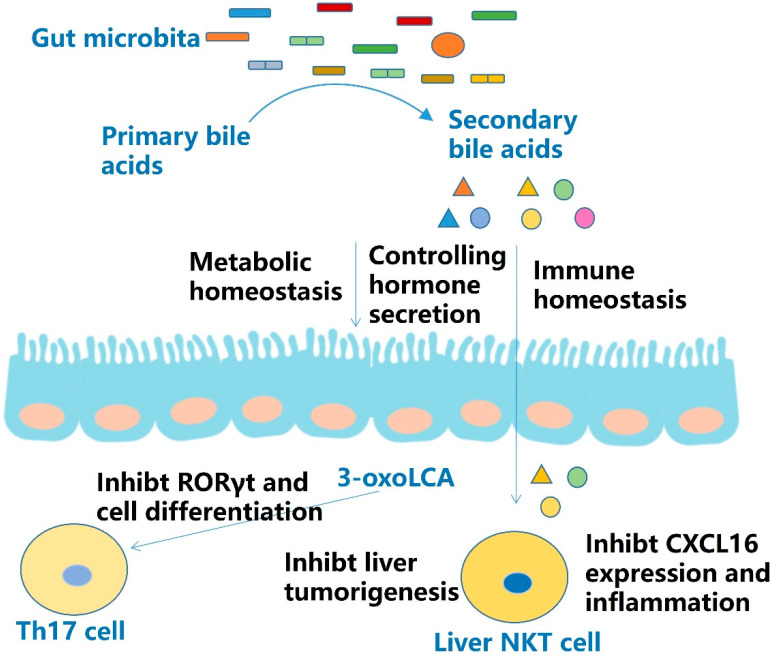
Effect and mechanism of secondary bile acids.

**Table 1 molecules-27-04799-t001:** Findings, mechanisms and producers of molecules.

Molecules	Findings	Mechanisms	Producers eg.
Propionate, butyrate, and valerate	Their contents in feces were significantly associated with longer progression-free survival in cancer patients receiving PD-1 mAb therapy [29].	The immune regulatory effects of SCFAs is dependent on HDAC, mTOR, GPRs and metabolic regulation. For example, SCFAs escalate the production of IL-22 by ILC3 and CD4^+^ T cells via GPR41, increase the cytotoxicity of CD8^+^ T cells by inhibiting HDAC activity, promote the expression of TNF-α and IFN-γ, and regulate T cells differentiation through mTOR pathway and energy metabolism.	SCFA-producing probiotics: *Clostridium butyricum, Bifidobacterium, Lactobacillus rhamnosus, Streptococcus thermophilus, Lactobacillus reuteri, Lactobacillus casei*, and *Lactobacillus acidophilus.* Propionate producer: *Bacteroidetes*, Butyrate producer: *Faecalibacterium*, *Eubacterium*, and *Roseburia.* For more see [33].
Propionate and butyrate	In NSCLC patients receiving second-line therapy with PD-1 mAb, propionate and butyrate concentrations were significantly associated with long-term beneficial effects [34].		
Butyrate	Serum butyrate and propionate were inversely correlated to PFS and OS in patients with metastatic melanoma receiving anti-CTLA-4 mAb treatmment. Butyrate reduced efficacy of CTLA-4 blockade in mice tumor models [35].	In patients, high concentrations of butyrate in the blood inhibited the accumulation of memory T cells and ICOS^+^ CD4^+^ T cells induced by anti-CTLA-4 mAb. In mice, butyrate inhibited upregulation of CD80/CD86 on dendritic cells and ICOS upregulation on T cells induced by CTLA-4 mAb, and increased the proportion of Treg cells [35].	
Valerate and butyrate	They enhanced the anti-tumor activity of antigen-specific CTLs and ROR1-targeting CAR-T cells in syngeneic murine melanoma and pancreatic cancer models [32].	Valerate and butyrate increase the function of mTOR as a central cellular metabolic sensor, and inhibit class I HDAC activity, resulting in elevated production of CD25, IFN-γ and TNF-α [32].	Butyrate producers: *Faecalibacterium prausnitzii* and *Anaerostipes hadrus*. Valarate producer: *Megasphaera elsdenii*. Valerate and butyrate producer: *Megasphaera massiliensis* [32].
Inosine	Inosine improved the efficacy of anti-CTLA-4 mAb and anti-PDL-1 mAb in mouse models with bladder cancer and small bowel cancer [36].	Inosine can regulate the Th1 differentiation and increase IFN-γ^+^ CD4^+^ and CD8^+^ T cell infiltration [36].	*Bifidobacterium pseudolongum* [36].
TMAO(TMA)	TMAO inhibited tumor growth and enhanced efficacy of anti-PD-1 mAb by activating CD8^+^ T cell-mediated antitumor immunity in mice [37].	TMAO improves anti-tumor immunity by activating PERK mediated ER stress, triggering Gasdermin-E mediated pyroptosis [37].	*Emergencia timonensis* and *Ihubacter massiliensis*…for more see [38].
Indoles	In murine pancreatic ductal adenocarcinoma models, indoles-derived tryptophan metabolites suppressed tumor immunity.	Indoles activate A2aR signaling in tumor associated macrophages, and inhibit the release of IFN-γ from infiltrating CD8^+^ T cells.	*Lactobacillus murinus* [39]. For more see [40].
Indole-3-lactic acid (ILA)	ILA suppresses intestinal tumorigenesis in mice [41].	ILA induces apoptosis of colon cancer cells via AhR [41].	*Lactobacillus gallinarum* [41]. For more see [42,43].
Indole-3-lactic acid	ILA promotes IL-22 expression during Th17 cell differentiation [44].	ILA acts on the AhR on the surface of T cells [44].	*Bifidobacterium bifidum*, *Bifidobacterium breve, Bifidobacterium longum**ssp. Longum, Bifidobacterium longum**ssp. Infantis and Bifidobacterium scardovii* [44].
Indole-3-propionic acid (IPA)	IPA reduces the progression of breast cancer in mice. IPA biosynthesis of the gut microbiota was inhibited in women newly diagnosed with breast cancer, especially stage 0 women [45].	IPA induces iNOS expression and enhanced mitochondrial reactive species production. IPA induces AMPK, FOXO1, and PGC1β, which are enzymes inducing mitochondrial biogenesis, in an AhR/PXR-dependent fashion [45].	*Providencia rettgeri* and *Alistipes shahii* [45]. *Clostridium botulinum. Clostridium paraputrificum, Clostridium cadvareris, Peptostreptococcus asscharolyticus, Peptostreptococcus russellii, Peptostreptococcus anaerobius*…for more see [42].
Indole-3-carboxaldehyde(3-IAld)	3-IAld can attenuate colitis induced by immunotherapy in mice, but does not affect therapeutic efficacy [46].	The beneficial activity of 3-IAld is achieved by increasing the intestinal barrier through the AhR/IL-22 axis and controlling inflammation through Treg cells [46].	*Lactobacillus acidophilus, Lactobacillus murinus, Lactobacillus reuteri* [47]. For more see [48].
Deoxycholic acid(DCA)	DCA promoted vasculogenic mimicry formation in intestinal carcinogenesis in mice [49].	Deoxycholic acid-induced proliferation and inhibited apoptosis in intestinal epithelial cells of Apc^min/+^ mice. DCA activated the vascular endothelial GFR2 signaling pathway to drive vasculogenic mimicry formation and the epithelial-mesenchymal transition process in mice [49].	*Clostridium* and *Bacteroides* [50].
3-oxolithocholic acid (3-oxoLCA) and isoLCA	They suppressed Th17 cell differentiation [51].	They inhibited ROR-γt, a key Th17-cell-promoting transcription factor [51].	*Gordonibacter pamelaeae P7-E3* and *Eggerthella lenta*. For more see [51].
Hippuric acid	The combination of hippuric acid, butyrylcarnitine, cysteine, and glutathione disulfide in plasma had a high response probability of PD-1 blockade therapy in NSCLC patients [52].	This may be due to their association with T cell metabolism [52].	*Clostridiales* and *Faecalibacterium prausnitzii* [53].
Muropeptides generated by *Enterococci*	It enhanced anti-PD-L1 mAb immunotherapy in mice B16-F10 tumor model [54].	Muropeptide exerted the immune activation function through the signal transduction of innate immune sensing protein NOD2, increased the proportion of CD8^+^ T cells in tumor infiltrating lymphocytes and expressing granzyme B [54].	*Enterococci faecium*…for more see [54].
Exopolysaccharide EPS-R1	It enhanced efficacy of anti-CTLA-4 mAb or anti-PD-1 mAb immunotherapy in CCL20-expressing tumor bearing mice [55].	Ingestion of EPS-R1 significantly increased the number of CCR6^+^ population in CD8^+^ tumor infiltrating lymphocytes (TILs). EPS-R1 further augmented the expression of IFN-γ as well as the genes encoding IFN-γ-inducible chemokines to enhance T cell function [55].	*Lactobacillus delbrueckii subsp. bulgaricus OLL1073R-1* [55].
FimH	It enhanced the effect of anti-PD-L1 mAb on CT26 xenografts in mice [56].	FimH can activate mouse and human NK cells by binding to TLR4 [57]. FimH combined with OVA can promote antigen-specific immune activation, promote T cell proliferation and IFN-γ and TNF-α production, increase the tumor infiltration of effector T cells and inhibit the growth of melanoma in mice. FimH in combination with anti-PD-L1 mAb can promote the activation of dendritic cells (DCs) [56].	Adhesin of type I pili on the surface of Gram-negative bacteria such as *Escherichia coli* [56].
Castalagin	It exerts antitumor activity and circumvents anti-PD-1 resistance through the gut microbiota [58].	Castalagin binds to the extracellular membrane of *Ruminococcus* and enriches for bacteria associated with efficient immunotherapeutic responses and improves the CD8^+^/FOXP3^+^CD4^+^ ratio within the tumor microenvironment. The effect is gut microbiota-dependent [58].	Polyphenol-rich berry camu-camu [58].

## Data Availability

Not applicable.

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
