# Peer review of "Effect of Gut Microbiota-Derived Metabolites on Immune Checkpoint Inhibitor Therapy: Enemy or Friend?"

_molecules, 2022, doi:10.3390/molecules27154799_

Round 1

Reviewer 1 Report

This review illustrates one of recent trends for cancer therapy, cancer immunotherapy. The review is interesting and focuses on the therapeutic/protective roles of some gut microbiota metabolites and related mechanisms. However, the review lacks a very important topic, during the review no specification to the most important microorganisms that play  crucial roles in producing the metabolites, postbiotics, contribute to improving immune cell functions and thus cancer therapy or protection. It is crucial to do a link between the microbial strains and their metabolites, by this way we can manipulate gut microflora in favor of the host health and improve gut eubiosis. Talking about the metabolites in general without mentioning these strains making the review very generic. Some metabolites such as phenolics are only mentioned in the introduction but their effects were not included in the review, however these groups of metabolites have very important effects on cancer immunotherapy and may be via different mode of actions, e.g. their antioxidant effects. Furthermore, a summary for studies on the effect of these metabolites on in vitro or in vivo cancer development should be shown. This can be done in table that contains the type of cancer cells, microorganism, metabolites, protective/therapeutic finding.

Specific comments:

Title, it must refer to the fact that the review focuses on immunotherapy for cancer. Furthermore, the question regarding the roles of microorganism : friend or enemy seems confusing. Through the article, they are friends?????

Key words: revise, there are only three key words and mostly are repatriations to some words in the title.

For all figures, show the full name of any short name (abbreviation) even if it was shown in the text.

Resolution of figure  two  is so bad and needs more clarification.

Line 206:  what does the abbreviation refer to?? 2.3. TMAO , avoid using abbreviations in the titles, subtitles and the banning of the sentence along the text.

Author Response

For the comments from Reviewer 1:

This review illustrates one of recent trends for cancer therapy, cancer immunotherapy. The review is interesting and focuses on the therapeutic/protective roles of some gut microbiota metabolites and related mechanisms. However, the review lacks a very important topic, during the review no specification to the most important microorganisms that play crucial roles in producing the metabolites, postbiotics, contribute to improving immune cell functions and thus cancer therapy or protection. It is crucial to do a link between the microbial strains and their metabolites, by this way we can manipulate gut microflora in favor of the host health and improve gut eubiosis. Talking about the metabolites in general without mentioning these strains making the review very generic. Some metabolites such as phenolics are only mentioned in the introduction but their effects were not included in the review, however these groups of metabolites have very important effects on cancer immunotherapy and may be via different mode of actions, e.g. their antioxidant effects. Furthermore, a summary for studies on the effect of these metabolites on in vitro or in vivo cancer development should be shown. This can be done in table that contains the type of cancer cells, microorganism, metabolites, protective/therapeutic finding.

Answer: Thank you for your valuable suggestions and comments. Since some metabolites were data derived directly by LC-MS analysis, there is insufficient evidence to support the microbiota from which they were derived. We tried our best to add the strains confirmed in the same study to the text. In addition, we read more literatures to find potential strains. We have made a table(Table 1), which contains the strains, anti-tumor effects and mechanisms of metabolites, as well as references.

There are few studies on phenols metabolized by microorganisms. We found only one study on immunotherapy: [52]Combination of host immune metabolic biomarkers for the PD-1 blockade cancer immunotherapy. JCI insight 2020. We have added it into the text(2.6 Other microbial-derived molecules).

Although the bioactivity of polyphenols from plants has been widely studied, there is only one related to immunotherapy: [58]A Natural Polyphenol Exerts Antitumor Activity and Circumvents Anti-PD-1 Resistance through Effects on the Gut Microbiota. Cancer Discov. 2022. We also added it to the text for its effect is gut microbiota dependent.

Specific comments:

  1. Title, it must refer to the fact that the review focuses on immunotherapy for cancer. Furthermore, the question regarding the roles of microorganism : friend or enemy seems confusing. Through the article, they are friends?????

Answer: Thanks for your comment. Among the metabolites of gut microbiota, some play a beneficial role in humans, such as SCFAs. SCFAs have not only been found to provide energy to colonocytes and maintain the intestinal mucosal barrier, but also reduce colon carcinogenesis and promote tumor immunotherapy efficacy.

  1. Key words: revise, there are only three key words and mostly are repatriations to some words in the title.

Answer: Thanks for your suggestion. We have added “Immune checkpoint inhibitor”, “host immunity” and “SCFAs” to the key words.

  1. For all figures, show the full name of any short name (abbreviation) even if it was shown in the text.

Answer: Thanks for your suggestion. But it is a common phenomenon that abbreviations appear in the figure. Many of the references we quoted can prove this. The abbreviations we used are common in international journals and meet the standards of journals. We were unable to revise all abbreviations because it will affect the impressions of figures.

  1. Resolution of figure two is so bad and needs more clarification.

Answer: Thanks for your suggestion. We have replaced it with a clear figure.

  1. Line 206: what does the abbreviation refer to?? 2.3. TMAO , avoid using abbreviations in the titles, subtitles and the banning of the sentence along the text.

Answer: Thanks for your suggestion. We have added the full name and checked the text.

Reviewer 2 Report

Review of “Effect of gut microbiota derived metabolites on immunotherapy: enemy or friend?“ (molecules-1794501)

This review focus on the effect of metabolites on the effect of immune checkpoint inhibitors and showed that the presence of metabolites has the influence of the effect of immune checkpoint inhibitors. The purpose of this review is interesting; however, several problems to be solved.

1.     This review was not a systematic review, but narrative review.

2.     The title included the “immunotherapy”, but the context of this study only focuses on the immune checkpoint inhibitors. How about the other immunotherapies?

3.     For each metabolite, it is desirable to summarize the mechanisms, data of animal model, and data of human model.

Author Response

Responses to the comments from Reviewer 2:

This review focus on the effect of metabolites on the effect of immune checkpoint inhibitors and showed that the presence of metabolites has the influence of the effect of immune checkpoint inhibitors. The purpose of this review is interesting; however, several problems to be solved.

  1. This review was not a systematic review, but narrative review.

Answer: Thanks for your suggestion. Due to many trials are in progress and there are small number of completed trials, we don't have sufficient data to make a systematic review or meta-analysis.

  1. The title included the “immunotherapy”, but the context of this study only focuses on the immune checkpoint inhibitors. How about the other immunotherapies?

Answer: Thanks for your suggestion. Since immune checkpoint inhibitors are currently the most widely used immunotherapy, related studies are naturally more numerous. There is only one study on CAR-T adoptive immunotherapy: [32]Microbial short-chain fatty acids modulate CD8(+) T cell responses and improve adoptive immunotherapy for cancer. Nat Commun 2021. We have added it to the text(2.1.1. SCFA enhance immunotherapy efficacy). In view of the emphasis of the manuscript, we revised the “immunotherapy” in title to “immune checkpoint inhibitor”.

  1. For each metabolite, it is desirable to summarize the mechanisms, data of animal model, and data of human model.

Answer: Thanks for your suggestion. We have made a table(Table 1), which contains the strains, anti-tumor effects and mechanisms of metabolites, as well as references.

Round 2

Reviewer 1 Report

The review article was modulated according to the recommendation. It is a piece of art and can be published in the journal.

Reviewer 2 Report

No further comments.